# Development of a core outcome set for lower limb orthopaedic surgical interventions in ambulant children and young people with cerebral palsy: a study protocol

Hajar Almoajil ![ORCID] ,[1,2] Helen Dawes,[3,4] Sally Hopewell,[1] Francine Toye,[5] Crispin Jenkinson,[6] Tim Theologis[1,7]

For numbered affiliations see end of article.

**Correspondence to**
Hajar Almoajil;
hajar.almoajil@ndorms.ox.ac.uk

## ABSTRACT

**Introduction** Musculoskeletal deformities and gait deviations are common features in ambulatory cerebral palsy (CP). Deformity correction through lower limb orthopaedic surgery is the standard form of care aimed at improving or preserving motor function. Current research on CP care does not always take into account individual patients' expectations and needs. There is a wide range of outcome domains and outcome measures used to assess outcome from treatment. This can lead to reporting bias and make it difficult to compare and contrast studies. A core outcome set (COS) would enhance the efficiency, relevance and overall quality of CP orthopaedic surgery research. The aim of this study is to establish a standardised COS for use in evaluating lower limb orthopaedic surgery for ambulatory children and young people with CP.

**Methods/analysis** A set of outcomes domains and outcome measures will be developed as follows: (1) a qualitative evidence synthesis to identify relevant outcomes from children and young people and family perspective; (2) a scoping review to identify relevant outcomes and outcome measures; (3) qualitative research to explore the experience of key stakeholders; (4) prioritisation of outcome domains will be achieved through a two-round Delphi process with key stakeholders; (5) a final COS will be developed at a consensus meeting with representation from key stakeholder groups.

**Ethics and dissemination** Ethical approval for this study was granted in the UK by the Oxfordshire Research Ethics Committee B (REC reference 19/SC/0357). Informed consent will be obtained from participants taking part in the qualitative research and Delphi process. Study findings will be published in an open access journal and presented at relevant national and international conferences. Charities and associations will be engaged to promote awareness of the project COS results.

**Trial registration number** COMET registration: 1236.
**PROSPERO registration number** CRD42018089538.

## Strengths and limitations of this study

► Rigorous core outcome set development methods that adhere to Core Outcome Measures in Effectiveness Trials guidelines.
► Different stakeholder inputs and engagement during the development of the protocol, in order to ensure outcomes that are important to patients, are identified.
► Support of several societies and charities that represent key stakeholders, which will facilitate participant recruitment during the Delphi process, dissemination and uptake of our core outcome sets.
► The Delphi exercise will be conducted using an online survey, limiting participants to those with access to the internet.

to three individuals per 1000 live births globally.[1 2] CP is defined as 'a group of permanent disorders of the development of movement and posture causing activity limitation[s] that are attributed to non-progressive disturbances that occurred in the developing fetal or infant brain'.[3] Functional mobility in children and young people with CP is usually classified according to the five-level gross motor function classification system (GMFCS), ranging from Level I, indicating maximal mobility with slight deficiencies in challenging activities, to Level V indicating full immobility and reliance on others.[4] Approximately two-thirds of the children are ambulant within GMFCS I, II, III.[5] Musculoskeletal deformities and resulting gait abnormalities are common and progressive during childhood, and lead to pathological and compensatory gait patterns.[6]

Many children with CP undergo lower limb orthopaedic surgery to address secondary musculoskeletal deformities and gait

## INTRODUCTION

Cerebral palsy (CP) is the most common cause of childhood physical disability, affecting two

abnormalities with the aim of improving or maintaining mobility.[7] Lower limb orthopaedic operations comprise a variety of soft tissue and bone procedures, including tendon/muscle lengthening or transfer and osteotomy or arthrodesis, often combined in the context of single-event multi-level surgery.[8–10]

Although many outcome measures for the evaluation of clinical trial results are in use, their application in the research context is largely inconsistent.[7] The consequences of research heterogeneity in outcome measures and outcome domains across studies limit the ability to compare findings among studies.[11 12] Generic and clinician-administered outcome measures such as clinical measurements of joint range of motion, spasticity, muscle strength and instrumented motion and gait analysis are employed in the majority of CP orthopaedic literature,[7] leaving the needs and expectations of both patient and caregivers unfulfilled.[7 13] Furthermore, several reviews in CP lower limb orthopaedic surgery clinical trials have been suspected of selective outcome reporting bias.[7 11 12] For example, McGinley and colleagues reported that adverse effects of single-event multi-level surgery weighed against its benefits are rarely addressed and only reported in a limited number of studies. Such inadequate reporting renders these studies less reliable and further reduces the ability to compare and combine studies through meta-analysis.[12]

The Core Outcome Measures in Effectiveness Trials (COMET) initiative brings together researchers interested in developing a standardised set of core outcomes in different health-related fields.[14] A core outcome set (COS) is defined as 'an agreed minimum set of outcomes that is recommended to be measured and reported in all clinical trials'.[15] In recent years, limited consensus has been reached about what outcomes should be measured in CP. For example, there are five International Classification of Functioning, Disability and Health (ICF) core sets for CP,[16–20] one comprehensive and four brief sets. The ICF core sets for children and young people with CP offer service providers and stakeholders an age-appropriate framework to explore functioning and disability for assessment, treatment, evaluation and policy purposes in a global context. Specifically, the ICF core sets for CP standardise what should be measured and reported—adopting the ICF biopsychosocial model. However, the ICF core sets do not include specific sets of interventions for this population.

Lower limb orthopaedic surgery is a common intervention in the management of ambulatory children with CP. However, significant variation in the outcomes collected and reported remains a challenge. The post-surgical emotional and physical challenge that this treatment imposes on children and their families is significantly bigger than any other gait improvement intervention in this population. It is therefore of the highest importance to ensure that the surgical aims and the expected outcomes are of relevance to children and their families. Developing a COS representing all stakeholders for this

specific intervention would represent the first step in this direction. Therefore, it is important to develop a COS that specifically addresses the needs of children and young people with CP undergoing lower limb orthopaedic surgery. A search of the COMET database and existing literature revealed that COS development has not been undertaken for the orthopaedic surgical management of lower limb problems in CP. Therefore, the aim of this study is to develop a COS for use in clinical trials involving lower limb orthopaedic surgery for ambulant children and young people with CP.

### Conceptual framework

The COMET initiative emphasises the need for a comprehensive framework of healthcare when developing a COS.[14] The WHO's International Classification of Functioning, Disability and Health for Children and Youth (ICF-CY) provides a useful framework that includes key aspects of a health condition and has been used extensively in CP research.[21–24] The ICF-CY has two parts: (1) functioning and disability and (2) contextual factors. Each part has two components; functioning and disability is subdivided into (a) body functions and structures and (b) activities and participation. Contextual factors are subdivided into environmental and personal factors.[25] Therefore, the ICF-CY framework taxonomy will be used as a basis of the development of COS.

### Scope

The scope of the COS will include four main areas based on the Core Outcome Set-STAndards for Development[26]: (1) population, (2) health condition, (3) intervention and (4) setting or context of use for which the COS is to be applied. Accordingly, the proposed COS will serve as an international standard that can be used to evaluate the overall success of lower limb orthopaedic surgery among ambulant children and young people with CP. It will comprise important outcomes (domains) and outcome measures that can be used in both clinical and research practice.

### METHODS AND ANALYSIS

#### Study design

The study design will follow the recommendations of the COMET initiative,[14] with reporting adhering to the Core Outcome Set-STAndardised Protocol Items statement.[27] COS development will involve two main parts, first, developing the COS domains and second, developing the core outcome measures. A brief overview of our study design, including estimated time frames, is highlighted in figure 1.

### Stakeholder involvement

Stakeholders are defined as those who provide insight regarding outcomes of importance during the development of the proposed COS. Box 1 lists potential stakeholders, which include healthcare professionals and

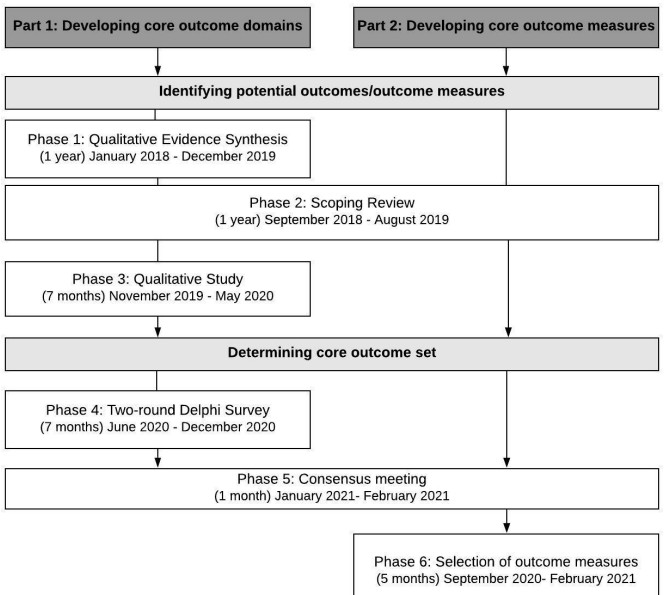

**Figure 1** Study design.

clinical academics (researchers), healthcare experts and children and young people with CP and their caregivers.

### Establishing a steering committee

An international steering committee will be formed to guide the development of this COS. The steering committee will include 10–15 individuals ranging from experts in orthopaedic interventions, researchers, parents and patient representatives. Committee members will be provided input through face-to-face meetings and email communications.

---

| **Box 1  Stakeholder involvement** |
| --- |
| **Clinical academics (researchers)**
► Healthcare scientists/researchers (expert in the fields of clinical research for cerebral palsy (CP), more specifically in the area of orthopaedics).
► Trialists and systematic reviewers.

**Healthcare professionals**
► Have at least 2 years of clinical experience in CP.
 – Orthopaedic surgeons.
 – Paediatrics physiotherapists, occupational therapists.
 – Paediatrics nurses.
 – Orthotists.
 – Clinical scientists.

**Children and young people, representative (family, carer)**
► Diagnosed with CP.
► Age from 8 to 18 years old at the time of the surgery.
► Ambulant or within level I, II, III of gross motor function classification system.
► Have undergone or are being considered for lower limb orthopaedic surgical intervention.
► Family or carer of a child or young people with earlier criteria. |

### Patient and public involvement

The development of this study protocol has been informed by discussions with a young adult with CP who has previously undergone lower limb orthopaedic surgery and with a parent representing approximately 40 families of children with CP. Representatives confirmed the importance of the topic and partnered in the design of the study and the information material to support the interview and the Delphi methods. As a result, the study design was refined, tips were given on how to maximise the sample of participants (caregivers and patients) and feedback was given on the participant information sheet for both the interview and Delphi studies and the interview topic guide questions.

## IDENTIFYING POTENTIAL OUTCOME DOMAINS AND OUTCOME MEASURES

### Phase 1: qualitative evidence synthesis

#### Research question and purpose

The review question for this study is: from the perspective of children with CP and their caregivers, what is the experience and expectation of outcomes after lower limb surgery?

The main purpose of the review is to (1) identify the aspect of health and outcomes that are considered important from the perspective of ambulatory children and young people with CP and their caregivers regarding to lower limb orthopaedic surgery and (2) explore how they experience lower limb orthopaedic surgery. A best-fit framework approach[28] will be used and the data from primary qualitative studies will be coded against an existing model. The ICF-CY framework[25] will be selected as a model 'best-fit' to extract the data from the included studies since the generated themes would represent a robust base of knowledge that form the first list of a future ICF-CY COS for lower limb orthopaedic surgery of ambulant children and young people with CP. Data that do not fit the ICF-CY will be analysed inductively into additional themes through thematic analysis.[29] Two reviewers will independently perform each stage of the review and resolve uncertainty through discussion. Review findings will be reported following enhancing transparency in reporting the synthesis of qualitative research statement guidelines to enhance transparency in reporting qualitative evidence synthesis.[30] The review protocol is available online via the PROSPERO database.

#### Search strategy

A systematic search will be conducted using four databases, Medical Literature Analysis and Retrival System Online (MEDLINE), Excerpta Medica Database (EMBASE), PsychINFO and Cumulative Index to Nursing and Allied Health Literature (CINAHL) with search terms and free-text terms combining four components: (1) CP; (2) perspective and experience; (3) orthopaedics, musculoskeletal and (4) qualitative research methodologies (online supplementary file 1). The search will include

relevant studies from inception to January 2018. In addition, the citation and the reference lists of all included studies will be checked through Web of Science citation search to identify any additional potentially relevant studies.

### Study selection and eligibility criteria

Potential studies will be exported into the EndNote X8.2 reference manager and duplicates will be removed. Studies will be put through a two-stage screening process. Titles and abstracts of studies identified by the search will be assessed against the inclusion criteria first; then, selected studies will be read in full to determine inclusion. Studies will be eligible if they meet the following inclusion criteria: (1) the study population consisted of individuals diagnosed with CP and/or their parent or primary caregiver; (2) the study participants were ambulatory or within level I, II, III of the GMFCS; (3) the study explored the experience of children and young people with CP and their family following lower limb surgery and their expectations of the surgery outcomes; (4) the study employed a qualitative design; (5) in the case of mixed-method studies, data from the quantitative and qualitative approaches were analysed and reported separately; (6) full article was published in English. Opinions, editorials and articles containing only quantitative data will be excluded. Studies with samples representing several types of disabilities will be excluded if the authors did not define findings by type or specify which findings were linked to individuals with CP. In addition, studies with only an abstract and those published as poster will be excluded as it would not be possible to adequately assess their methodological quality.

### Quality appraisal

Identified studies will be assessed by the critical appraisal skills programme tool.[31] Studies will also be assessed using the Dixon-Woods and colleagues categorisation method,[32] which consists of a global appraisal of whether the study was a key or satisfactory paper, as opposed to a paper that was irrelevant to the synthesis or methodologically fatally flawed.

### Data extraction

A template will be created for extracting data from the identified studies. The major elements comprised key research aims; details of the research context; participants; data collection and analysis methods.

### Synthesis

In order to map data onto precise ICF-CY codes, the ICF linking rule will be used (table 1).[33] Each line of text will be coded according to its meaning and will be linked to the most precise ICF-CY code. If the content of a code is not explicitly named in the ICF-CY category, the 'not defined' and 'not covered' category will be applied. Data that do not fit the ICF-CY will be analysed thematically.[29]

The data will be linked to the ICF-CY separately by two reviewers. A third independent reviewer with expertise in the concepts and taxonomy of the ICF-CY will be consulted to resolve disagreements between the two reviewers concerning the selected categories. The process will be discussed and refined by three reviewers to develop a final list of relevant themes.

### Phase 2: Scoping review
#### Research question and purpose

This review aimed to answer these questions: (1) what outcomes are reported in the medical literature after

| Table 1 | ICF linking rules |
| --- | --- |
| Number | Rule |
| 1 | Acquire good knowledge of the conceptual and taxonomical fundamentals of the ICF, as well as of the chapters, domains and categories of the detailed classification, including definitions before starting to link meaningful concepts to the ICF categories |
| 2 | Identify the main concept(s) most relevant to be linked to the ICF |
| 3 | Identify any additional concepts contained in the piece of information in addition to the main concept(s) already identified in the previous step |
| 4 | Identify and document the perspective taken on within a certain piece of information when linking it to the ICF |
| 5 | Identify and document the categorisation of the response options |
| 6 | Link all meaningful concepts, the most relevant and additional ones, to the most precise ICF category |
| 7 | Use 'other specified' or 'unspecified' ICF categories as appropriate |
| 8 | If the information provided by the meaningful concept is not sufficient for making a decision about the most precise ICF category, assign the concept to nd (not definable) |
| 9 | If the meaningful concept is not contained in the ICF, but is clearly a personal factor as defined in the ICF, assign the meaningful concept to pf (personal factors) |
| 10 | If the meaningful concept is not contained in the ICF, assign this meaningful concept to nc (not covered) |

ICF, International Classification of Functioning, Disability and Health.

lower limb orthopaedic surgery for ambulant children with CP? And (2) what outcome measures are used in this field?

The purpose of this scoping review is to identify relevant outcome domains and outcome measures used in clinical trials of lower limb orthopaedic surgery for CP. Although previous scoping reviews for studies published between 1990 and 2015 were identified,[7 34] updating the review was important to ensure that recently published outcome domains and outcome measures were identified. This was important particularly because, in recent years, researchers and healthcare professionals have become more aware of patient priorities and have acknowledged the value of patient-reported outcomes. These outcomes and outcome measures will underpin the consensus on selecting outcomes of important and outcome measures. According to the Cochrane Review recommendation, a review update should be re-conducted using the same methods as the original review[35]; therefore, this review was in line with the methodology of the original scoping review.[7] The review will be developed in accordance with the Preferred Reporting Items for Systematic Reviews and Meta-Analyses Protocol guidelines. Two reviewers will independently undertake each stage of the review. The research team will resolve any uncertainty and provide consensus after each stage through discussion. The review will be reported following the Preferred Reporting Items for Systematic review and Meta-Analysis for Scoping Reviews (PRISMA-ScR) checklist.[36]

### Search strategy

A systematic search of five databases (MEDLINE, PubMed, EMBASE, CINAHL and the Cochrane Controlled-Trials Registry) will be performed. Search terms including 'cerebral palsy' AND 'surgical procedures' OR 'surgery' OR 'operative' will be used. Reference lists of all studies identified will be hand searched to identify any additional potentially relevant studies. All studies published between January 2016 and June 2019 will be included.

### Study selection

Potential studies will be exported into the reference manager, EndNote X8.2, and duplicated studies will be removed. Studies will then go through a two-stage screening process. First, the titles and abstracts will be assessed against the inclusion criteria. Second, a full-text screening will be undertaken of all studies that were identified in the first step.

### Eligibility criteria

Studies will be eligible if they meet the following inclusion criteria: (1) ambulatory children and young people diagnosed with CP; (2) participants are aged between 0 and 20 years old; (3) have had lower limb orthopaedic surgery; (4) studies reported at least one outcome measure; (5) each outcome measure must have at least one published paper reporting its psychometric properties; (6) the full article was published in English. Purely observational

investigations and qualitative studies, grey literature, studies involving only adults, those with patients receiving alternative therapy or pharmacologic interventions, and papers that reported surgery performed only for hip dysplasia will be excluded.

### Data extraction

A pre-designed standardised data-extraction form will be used to extract the main characteristics of identified studies. Information extracted will include author, year of publication, aim/purpose, study population and sample size, methodology, intervention type, outcome domains and outcome measures used.

### Analysis

Content analysis will be used to identify the breadth of content of the identified outcome domains and measures, which will then be mapped to the ICF-CY domains. After mapping, each outcome domain and measure will be analysed descriptively. Descriptive analyses will be conducted to determine the frequency and proportion of each outcome reported and of instruments used to assess identified outcome domains.

## Phase 3: qualitative study

### Research question and purpose

The following question will guide the study: what outcome domains are considered important by ambulant children and young people with CP, their caregivers and health professionals after lower limb orthopaedic surgery? Children and young people with CP and their families may prioritise different outcomes post-surgery. Thus, the aim of the qualitative study is to identify important outcome domains from each stakeholders' group.

### Study sample

A purposive sampling strategy[37] will be used to include a range of stakeholders: (1) children and young people with CP and their representatives (family, carer); (2) health professionals who have at least 2 years of clinical experience in CP (box 1). The study will only include English speakers who have the capacity to give informed consent for themselves or their children (under 16 years old). Data saturation will be defined as a point at which data from three consecutive interviews do not contribute to additional themes. It is likely that a sample of 20 participants will allow data saturation.[38]

### Recruitment

Participants will be recruited from the paediatric orthopaedic clinic at the Nuffield Orthopaedic Centre in Oxford. Information for potential participants will be provided verbally by the usual care team and a participant information leaflet will be used. Healthcare professionals and clinical academics (researchers) will be invited through the usual care team using their publically available professional details.

## Consent

Written informed consent will be obtained from potential participants prior to conducting the interviews. Individuals who do not have capacity to give informed consent will not be included in the study. It will be stressed that participants are under no obligation to take part and they are free to withdraw at any time without affecting their medical care (CP children and their representatives) or legal rights (for healthcare professionals and researchers).

## Data collection

Semi-structured interviews will take place at a time and place convenient to the participant. This might be in a clinic room at the hospital or at the participant's home. Young adults (16–18 years old) will be given the option of an interview separately from their caregivers. The following demographic information will be collected: child age, identity, operation history, year of health professional's experience. Interviews will be semi-structured and include several open-ended questions to encourage participants' thoughts and opinions, guided by a topic guide to ensure key areas are covered. The interview topic guide will be formed and shaped by the findings of phases 1 and 2. For example, participants will be asked to reflect on the identified outcome domains from the reviews. Child-friendly techniques through using a talking mat (https://www.talkingmats.com/) will be available for children aged 8–15 years old to enhance children's participation and interest in answering the questions. Talking mat is an interactive resource that designed picture symbols to facilitate communication and provide structured framework for open questions. To facilitate children's participation and engagement, a play specialist will be available during the children interviews.

## Data analysis

Interviews will be audio-recorded and transcribed. Participant will be identified only by a code (participant ID number) on trial documents and in any electronic database to maintain confidentiality.

Data will be analysed through content analysis. Data will be coded, indexed and charted to identify themes or patterns of key points and priorities.[39] The list of prioritised outcomes generated by the analysis will be systematically classified using an international ICF framework linking process outlined by Cieza and colleagues.[33] If participants identify domains that are outside the ICF categories, these will be documented, analysed and reported. The resulting ICF codes will be analysed in terms of their representation across ICF-CY components and between stakeholder groups.

## Potential outcome domains and outcome measures

A list of potential outcomes domains derived from the qualitative evidence synthesis (phase 1), scoping review (phase 2) and semi-structured interviews (phase 3) will be collected and assessed by the COS steering committee.

The steering committee will revise the phrasing of the potential domains to confirm clarity, relevance and suitability prior to the Delphi process (phase 4).

# DETERMINING CORE OUTCOME SETS
## Phase 4: Delphi process
### Research question and purpose

The study aims to answer the following question: which outcomes do children and young people with CP, caregivers and healthcare professionals think should be included in a COS for lower limb orthopaedic surgery? The aim of the Delphi survey[40] is to gain consensus on important lower limb orthopaedic surgery outcome domains from the perspective of stakeholders.

### Selection of panel members and sample size

The eligibility criteria shown in Box 1 will be used as a guide in the selection of the Delphi panel members. Potential Delphi panel members will be invited to a two-stage Delphi survey. The aim is to maintain a minimum of 40 participants representing each stakeholder group throughout the Delphi rounds, including (1) children and young people and their caregivers, and (2) health professionals. Balanced representation of multiple viewpoints, and expertise will be considered.[41] Based on an estimated attrition of 30% across rounds,[14] the initial target recruitment will be approximately 100 participants.

### Recruitment

Different strategies will be followed to identify the potential panel, as shown in box 2. Clinical academics and healthcare professionals will be invited through professional societies. Specific invitations to authors of relevant

---

**Box 2 Selection of Delphi panel member**

**Clinical academics (researchers)**
► Personalised recruitment emails.
  – Researchers of primary studies identified from the scoping review.
  – Healthcare scientists/researchers (expert in the fields of clinical research for cerebral palsy (CP), more specifically in the area of orthopaedics).
  – Trialists and systematic reviewers.
► Snowball sampling.

**Healthcare professionals**
► Personalised recruitment emails.
  – National and international societies and associations focusing on paediatric CP.
► Social media of these societies, associations and organisations.
► Snowball sampling.

**Children and young people, representative (family, carer)**
► Personalised recruitment emails.
  – National and international charities of CP, disabilities, families.
  – Patients/families attending clinics at the Nuffield Orthopaedic Centre, Oxford.
► Social media of the earlier charities.

---

references identified through the qualitative evidence synthesis and scoping review will be also targeted. Snowballing techniques will be used to ensure a representative sample of international researchers and clinicians are invited. Children and family recruitment will be initiated through the clinical care team at Nuffield Orthopaedic Centre (Oxford, UK) and will expand nationally and internationally to include patient and parent organisations and charities.

### Delphi survey

Adapted participant information leaflets developed from the COMET initiative will be used to outline the rationale for the development of the proposed COS and describe the requirements for taking part in the Delphi. A child-friendly animation will be developed to explain the rationale of the study and promote children's understanding of the purpose of the COS (online supplementary file 2). The Jisc online survey tool will be used. The Delphi survey will be pilot tested by the members of the steering committee to assess face validity, understanding and acceptability. Demographic information will be collected at the start of the Delphi survey to describe the study sample, and to provide each respondent with a unique identifier enabling personalised reminders for completion of subsequent rounds. This information will include the stakeholder group that the participant belongs to, the age group, years of experience, and position.

Participants will be asked to score each outcome in the Delphi using a GRADE scale, which ranges from 1 to 9 (1 to 3=not important, 4 to 6=important but not critical and 7 to 9=critical for inclusion).[42] The proposed scoring system was selected to facilitate maximum discrimination between items. There will be free-text fields to allow the participants to give a reason for their decision and/or any additional outcomes that they consider to be important. Items scored between 7 and 9 (critical importance) by ≥75% will be directly moved to the following stage (ie, consensus meeting) for discussion. Items scored between 1 and 3 (not important) by ≥25% will be excluded from the round 2 of the Delphi survey. Only items scored between 4 and 6 (important but not critical) will be carried out in round 2 of the Delphi survey. Additional outcomes from the free-text field will be reviewed by the research team and where appropriate carried forward to round 2. During round 2 of the Delphi survey, participants will be asked to re-score the importance of each outcome that were scored between 4–6. A consensus process will be carried out similar to that in round 1. A report with findings including all items results and consensus category will be presented in the consensus meeting. Each round will be open to the panel for 4 weeks and reminder emails will be sent at 2 week intervals in order to maximise follow-up rates. Only the research team will have access to the complete list of those taking part in the Delphi survey.

### Analysis

Descriptive statistics will be undertaken using Statistical Package for the Social Sciences (SPSS) software to summarise the distribution of scores and to calculate the median and IQR for each Delphi survey item. The denominator for each Delphi survey item will be the number of participants completing that item, rather than the number of participants completing the Delphi survey overall (ie, a participant may choose not to score a particular Delphi item for whatever reason). Survey data will be analysed separately for each stakeholder group. Outcomes from both rounds will be analysed descriptively; the number of participants rating each outcome from rounds 1 and 2 will be calculated.

### Phase 5: consensus meeting
#### Recruitment
A consensus meeting will be hosted for the purpose of finalising the COS. The meeting will include approximately 20 panel members. Representatives from all stakeholder groups, representing as much geographical, ethnic, demographic and cultural diversity as possible (as recommended by COMET) will be invited at this stage. Ten of those participants will be randomly selected from the Delphi survey participants and the study steering group. Face-to-face and remote access to the meeting will be available.

#### Final decisions
At the meeting, Delphi survey results will serve as the basis for the discussion and development of the final COS to be agreed across stakeholder groups. Across all stakeholder groups, any outcome categorised as 'consensus in' will be proposed to be included in the final COS, while any outcome categorised as 'consensus out' will be excluded. The panel members will electronically vote to accept this proposal or suggest outcomes that warrant further discussion.

Outcomes that are differently categorised by different stakeholder groups and those categorised as 'no-consensus' will be discussed individually. A second round of voting will be used to agree the final COS. A second meeting will be arranged in the event of no agreement on the final COS at the end of the first meeting. Based on COMET recommendations, the final COS is expected to include five to ten outcomes.[14]

### Phase 6: selection of outcome measure
After the development of a COS, it is recommended to identify a set of measurements, the 'outcome measures' that would be used to evaluate the selected outcomes.[43] In order to establish a core outcome measures set, a four-step process will be followed: (1) conceptual considerations (scope); (2) identifying existing outcome measures; (3) quality assessment of the identified outcome measures and (4) generic recommendations for the selection of outcome measures for a COS.

The conceptual considerations of the proposed core outcome measures set will be associated with the study scope. Accordingly, all available outcome measures used in clinical research following lower limb orthopaedic surgery in ambulant CP will be considered for the core set. An international perspective on the subject will be captured by involving stakeholders from the study's international steering committee and consensus panel members.

Previous reviews on this field[7 34] alongside the scoping review (phase 2) will be used as a starting point to identify currently used outcome measures. The quality assessment of each outcome measure identified will be determined by the available systematic reviews using the consensus-based standards for selection of health measurement instruments (COSMIN). This will assess the psychometric properties of outcome measures that have been used in CP clinical studies. For example, two recent systematic reviews using a modified COSMIN method to assess gait-related outcome measures in CP will be employed to choose suitable, high-quality outcome measures.[44 45]

A consensus meeting with a panel of health professionals will subsequently be organised to establish appropriate outcome measures for each outcome domain identified during the COS development. Members will be asked to recommend one high-quality outcome measure per core outcome domain. If no adequate outcome measures exist for a specific core outcome, this will be acknowledged, and recommendation will be made for future development of an adequate high-quality outcome measure.

## ETHICS AND DISSEMINATION

Informed consent will be obtained from all participants taking part in the interview and Delphi process. All procedures will be conducted according to the Declaration of Helsinki.

Support of societies, associations and charities that represent health professionals, families of children and young people with CP will facilitate dissemination of the COS and subsequent uptake, for example, the 'British Society for Children Orthopaedic Surgery', 'Step charity' and 'Action Cerebral Palsy'. A one-page summary will be provided to the clinicians and researchers and a lay language summary will be available for the patients and their caregivers to the relevant societies. The findings will be submitted for publication in peer-reviewed and open access journals and will be presented at national and international conferences on CP. Journals and funding bodies will be approached to promote awareness of the COS results.

## DISCUSSION

To our knowledge, this is the first study on the development of a COS for lower limb orthopaedic surgery in ambulant children with CP. This study employed a well-established and widely used method developed by the COMET Initiative. Involving patients in COS development has become common practice to ensure the relevance of the proposed COS to all stakeholders. In this protocol, children and families will be directly engaged with the COS development through participation in the steering committee, the interviews, the Delphi process and the consensus meeting.

This study includes a comprehensive search for potentially relevant outcomes through qualitative evidence synthesis, a scoping review and interviews with stakeholders' groups. This process will be conducted by at least two independent researchers ensuring identification of all potential outcomes. This will provide a comprehensive list of all pertinent outcomes for the Delphi survey.

As the comprehensive search for outcomes focuses on the English literature and on interviews with English speaking stakeholders, any outcomes available in the non-English literature may be omitted. Free-text fields will be included in the Delphi survey to allow participants to suggest any additional outcomes that they consider important. The potential imbalance between national and international participants may represent a limitation of the study. Although the proposed COS development will aim to reach international consensus, it is possible that most participants will be recruited from the UK, which may affect the wider generalisability of the COS findings.

**Author affiliations**

[1]Nuffield Department of Orthopaedics, Rheumatology and Musculoskeletal Sciences, University of Oxford, Oxford, UK

[2]Department of Physical Therapy, College of Applied Medical Science, Imam Abdulrahman Bin Faisal University, Dammam, Saudi Arabia

[3]Centre for Movement, Occupational and Rehabilitation Sciences, Faculty of Health and Life Sciences, Oxford Brookes University, Oxford, UK

[4]Department of Clinical Neurology, University of Oxford, Oxford, UK

[5]Physiotherapy Research Unit, Nuffield Orthopaedic Centre, Oxford University Hospitals NHS Trust, Oxford, UK

[6]Nuffield Department of Population Health, University of Oxford, Oxford, UK

[7]Paediatric Orthopaedic Surgery, Nuffield Orthopaedic Centre, Oxford University Hospitals NHS Foundation Trust, Oxford, UK

**Contributors** All authors contributed to the design of this protocol. The study protocol was developed by HA, HD, SH, FT and TT. The manuscript was drafted by HA and was refined by all authors (HD, SH, FT, CJ and TT). TT is responsible for the management of the study and is the principal investigator for the study. HD, SH and FT provide supervision and have had input to all aspects of the study. CJ advised on the design of the protocol. All authors edited the manuscript and read and approved the final version.

**Funding** This study is funded by the Imam Abdulrahman Bin Faisal University, Saudi Arabia as part of HA's PhD program. HD is supported by the Elizabeth Casson Trust and the National Institute for Health Research (NIHR) Oxford Health Biomedical Research Centre. TT was funded by the Oxford Biomedical Research Centre.

**Competing interests** None declared.

**Patient consent for publication** Not required.

**Provenance and peer review** Not commissioned; externally peer reviewed.

**ORCID iD**

Hajar Almoajil http://orcid.org/0000-0001-5308-3362

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
