## [Reviewer comments · BMJ Open]

ARTICLE DETAILS

TITLE (PROVISIONAL)	Development of a core outcome set for lower limb orthopaedic surgical interventions in ambulant children and young people with cerebral palsy: a study protocol
AUTHORS	Almoajil, Hajar; Dawes, Helen; Hopewell, Sally; Toye, Francine; Jenkinson, Crispin; Theologis, Tim

VERSION 1 - REVIEW

REVIEWER	Verónica Schiariti Division of Medical Sciences, University of Victoria, Canada
REVIEW RETURNED	18-Oct-2019

GENERAL COMMENTS	Thank you for the opportunity to review this protocol entitled Development of a core outcome set for lower limb orthopaedic surgical interventions in ambulant children and young people with cerebral palsy: a study protocol. The aim of this study is to establish a standardised COS for use in evaluating lower-limb orthopaedic surgery for ambulatory children and young people with CP. I commend the authors for seeking to publish the protocol. The availability of a COS protocol would ensure that the methods are explicitly documented before the COS development project starts and that the COS-STAP minimum standards are followed. Before considering this protocol for publication, the following revisions are needed: INTRODUCTION Third paragraph • Correct typo "...limited consensus has been reached about what outcomes should bmeasured in CP"• Change the "International Classification of Function (ICF)" to its complete name of the classification International Classification of Functioning, Disability and Health (ICF)• Section referring to the ICF Core Sets for CP needs revisions. This sentence "However, this core set is a generic one for the CP population and does not specifically address specific interventions in specific population groups" does not reflect the purpose of the ICF Core Sets for CP. I suggest the following: There are five ICF Core Sets for CP, one comprehensive and four brief sets. The ICF Core Sets for children and young people with CP offer service providers and stakeholders an age-appropriate framework to explore functioning and disability for assessment, treatment, evaluation, and policy purposes in a global context. Specifically, the ICF Core Sets for CP standardize what should be
---

measured and reported – adopting the ICF biopsychosocial model. However, the ICF Core Sets do not include specific sets of interventions for this population.

- The last sentence referring to the GMFCS seems out of place. The purpose of this classification system is to describe the gross motor function of children and youth with cerebral palsy based on their self-initiated movement - mainly sitting, walking, and wheeled mobility. The GMFCS does not address outcomes or what to measure in CP.

Use CP consistently throughout the manuscripts, e.g. last paragraph of introduction

CONCEPTUAL FRAMEWORK

- Provide references here ... The ICF-CY ...has been used extensively in CP research. (REFERENCES)
- The ICF has two parts with two components each, please revise this statement “The framework provides three main domains: body function, activity, and participation”

- Correct typo CF-CY

METHODS AND ANALYSIS

Study design

- Please add an estimated timeline for the completion of both phases

Establishing a steering committee

- As the output of the project is to develop international standards, are you considering international representation of members in the steering committee?
- Also, consider moving the information regarding the phases of the study under this subheading to a different subheading as the information does not belong here

Phase 1 – systematic review of domains

- What is the research question for this systematic review?
- Study selection and criteria: the following criteria needs further clarification “(3) the study explored the experience of children and young people with CP and their family of lower limb orthopaedic surgery” it is not clear at this point what outcomes are you looking for, the experience during the surgical period?, after surgery?, the outcomes of a surgical intervention? Measured using a standardized tool/s? Themes related to goals for surgical interventions?

- ICF linking rules – you might want to use the revised version of the linking rules published in March 2019 published in <https://www.ncbi.nlm.nih.gov/pubmed/26984720> You will need to update Table 2, there are 10 rules proposed now.

Phase 2 – Scoping review

- Search strategy needs revisions as key terms are missing, for example the search terms described in this section do not include the term “OUTCOME”, or tools or measures, or “GAIT”, you should maximize the search strategy, you might want to consultant an experienced librarian to help with this task due to the importance of this step

- Can you provide a rationale for conducting two systematic reviews, you could run a search for domains and outcomes including qualitative and quantitative studies.

Phase 3 – Qualitative study

- What is the research question that you want to answer with this study?

	 • Inclusion criteria: you might want to add a criteria that include “lower limb surgical intervention” as you are creating a COS for this population Phase 4 – Delphi process  • Participants: you are inviting children to participate in this two-stage scoring exercise, how are you planning to present the information to this group? • Also, the output of this project is meant to be international, are you planning to include international stakeholders? The recruitment strategy is not described for this study. Phase 5 – Consensus meeting  • Recruitment strategy of panel members? • International representation? • Describe the procedure for determining how outcomes will be added/combined/dropped from consideration during the consensus process • Describe how missing data will be handled during the consensus process Additional comments:  • Include potential limitations to the proposed studies and successful completion of proposed project • Timeline of the total project • Figure 1. Please check misspellings in the main boxes • Sources of funding? • ICF-CY used in first study, are you planning to use the ICF in the other studies as well?
--	---

REVIEWER	Toby Smith University of East Anglia, UK I have a position within NDORMS, University of Oxford, UK.
REVIEW RETURNED	08-Jan-2020

GENERAL COMMENTS	Abstract – clear presentation of the approach taken. Registration with COMET was acknowledged. Strengths and limitations – appropriate and well-reasoned. Introduction – Population clearly defined and sufficient context provided. Acknowledgement of merits of Core Outcome Sets on research synthesis and efficiency justified. Further clarification on why this core outcome set is warranted over that cited in Page 4-5 would be beneficial. There is suggestion of a requirement for one related to ‘motor function’ however this message could be more clearly articulated. I am not convinced based on the current text why focusing on lower-limb orthopaedic surgery is required and the current text questions whether this is too narrow and the same COS would be relevant for pre- and post-surgery i.e. those with motor challenges. Based on the stage of this study, I would suggest that revising the text to clarify this would be beneficial. This has the potential to be really valuable and therefore please clarify/sell-this further to the reader.
---

	Methods – reporting by COS-STAP which is appreciated. Wider stakeholder group which is a huge benefit. Overall the methods are well-presented, clearly documenting and justifying the approaches taken. I am unsure about the merit of Table 2. Could this not just be cited rather than presented in the text? Justification for the narrow search period (Jan 2016 to June 2019) would be helpful for the scoping review? Why has this been undertaken when important studies may be available before 2016? There is limited feed-forward of the work packages into the qualitative Phase 3 study. I would suggest that these findings from Phase 1 and 2 may be helpful to inform the topic guides for Phase 3. Should this be considered? Further detail is required in Table 1 as to who the Health Professionals are. Could the team please provide the professions they expect to approach i.e. surgeons, play therapists, physiotherapists, occupational therapists, orthotists etc etc. This would be really helpful. The Phase 4 Delphi is appropriately reported however please provide further information on who will be completing this, where it will be sent to, who the target recruitment sites are, what approaches are made to improve the international reach of the Delphi. This is important. Phase 5 needs further clarification. Phase 4 will inform what domains should be included. However the matching of domains to outcomes is a challenging process and the current text requires further detail. There is a suggestion that outcomes previously associated with the selected domains will be linked and their feasibility/psychometric properties will be evaluated against the COSMIN checklist – this process warrants its own Phase in reporting so the reader can appreciate how this is actually undertaken. There is currently insufficient information here. Furthermore, where will the stakeholder meetings take place. Are these all in Oxford? How generalisable/transferable are the views of this group? Ethics and Dissemination – appropriate Limitations – there is currently insufficient acknowledgement of the issue of generalisability/transferability of the COS internationally. The searches are based on English language publications. The consensus meetings are ‘presumably’ in Oxford, the Delphi reach feels UK base (but may not be). COS should be international in their importance and relevance. The current approach feels very Oxford-centric. This should be acknowledged somewhere and/or strategies highlighted to the reader as to where this has been negated.
--	--

VERSION 1 – AUTHOR RESPONSE

Reviewer 1
Introduction

Comment 1: Correct typo “...limited consensus has been reached about what outcomes should bmeasured in CP”

Response: We have corrected the typo in the revised manuscript.

'In recent years, limited consensus has been reached about what outcomes should be measured in CP' p.4

Comment 2: Change the "International Classification of Function (ICF)" to its complete name of the classification International Classification of Functioning, Disability and Health (ICF)

Response: We have changed it to be in full classification name.

'only one comprehensive International Classification of Functioning, Disability and Health (ICF) core outcome set was developed' p.4

Comment 3: Section referring to the ICF Core Sets for CP needs revisions. This sentence "However, this core set is a generic one for the CP population and does not specifically address specific interventions in specific population groups" does not reflect the purpose of the ICF Core Sets for CP. I suggest the following:

There are five ICF Core Sets for CP, one comprehensive and four brief sets. The ICF Core Sets for children and young people with CP offer service providers and stakeholders an age-appropriate framework to explore functioning and disability for assessment, treatment, evaluation, and policy purposes in a global context. Specifically, the ICF Core Sets for CP standardize what should be measured and reported – adopting the ICF biopsychosocial model. However, the ICF Core Sets do not include specific sets of interventions for this population.

Response: Thank you for this, we have adjusted the text as suggested.

'There are five ICF Core Sets for CP, one comprehensive and four brief sets. The ICF Core Sets for children and young people with CP offer service providers and stakeholders an age-appropriate framework to explore functioning and disability for assessment, treatment, evaluation, and policy purposes in a global context. Specifically, the ICF Core Sets for CP standardize what should be measured and reported – adopting the ICF biopsychosocial model. However, the ICF Core Sets do not include specific sets of interventions for this population.' p.4-5

Comment 4: The last sentence referring to the GMFCS seems out of place. The purpose of this classification system is to describe the gross motor function of children and youth with cerebral palsy based on their self-initiated movement - mainly sitting, walking, and wheeled mobility. The GMFCS does not address outcomes or what to measure in CP.

Response: We agree with this point and the text was deleted accordingly.

'Since CP is a heterogeneous condition, a classification based on motor function was recommended to better define clinical groups based on lower limb functional ability (e.g., Gross Motor Function Classification System).' Deleted since its 'out of place'

Comment 5: Use CP consistently throughout the manuscripts, e.g. last paragraph of introduction

Response: Cerebral Palsy changed to CP

'Therefore, the aim of this study is to develop a core outcome set for use in clinical trials involving lower limb orthopaedic surgery for ambulant children and young people with CP' p.5

Conceptual Framework

Comment 6: Provide references here ... The ICF-CY ...has been used extensively in CP research. (REFERENCES)

Response: Four references added to the sentence.

'The World Health Organization' International Classification of Functioning, Disability and Health - Children and Youth (ICF-CY) provides a useful framework that includes key aspects of a health condition and has been used extensively in CP research.²¹⁻²⁴ p.5

21. Burak, M, Kavlak, E. Investigation of the relationship between quality of life, activity-participation and environmental factors in adolescents with cerebral palsy. *NeuroRehabilitation*. 2019(Preprint):1-11.
22. Geijen, M, Ketelaar, M, Sakzewski, L, et al. Defining Functional Therapy in Research Involving Children with Cerebral Palsy: A Systematic Review. *Physical & Occupational Therapy In Pediatrics*. 2019:1-16. 10.1080/01942638.2019.1664703
23. Lidman, GRM, Nachemson, AK, Peny-Dahlstrand, MB, et al. Long-term effects of repeated botulinum neurotoxin A, bimanual training, and splinting in young children with cerebral palsy. *Developmental Medicine & Child Neurology*. 2020;62(2):252-258. 10.1111/dmcn.14298
24. Schiariti, V, Oberlander, TF. Evaluating pain in cerebral palsy: comparing assessment tools using the International Classification of Functioning, Disability and Health. *Disability and Rehabilitation*. 2019;41(22):2622-2629. 10.1080/09638288.2018.1472818

Comment 7: The ICF has two parts with two components each, please revise this statement “The framework provides three main domains: body function, activity, and participation”

Response: This section has been revised in the protocol.

'The ICF-CY has two parts: 1. functioning and disability and 2. contextual factors. Each part has two components; Functioning and Disability is subdivided into a. Body Functions and Structures and b. Activities and Participation. Contextual Factors are subdivided into Environmental and Personal Factors. p.5-6

Comment 8: Correct typo CF-CY

Response: We have corrected the typo in the revised manuscript.

'Therefore, the ICF-CY framework taxonomy will be used as a basis of the development of COS.' p.6

Methods and Analysis

Comment 9: Study design, please add an estimated timeline for the completion of both phases

Response: We have added the timeline for each phase in figure 1 and highlighted in text under 'study design' section

'A brief overview of our study design, including estimated time frames, is highlighted in Figure. 1.' p.6

Comment 10: Establishing a steering committee, As the output of the project is to develop international standards, are you considering international representation of members in the steering committee?

Response: An International steering committee will be considered
'An international steering committee will be formed' p.7

Comment 11: Also, consider moving the information regarding the phases of the study under this subheading to a different subheading as the information does not belong here

Response: Information regarding the phases of the study was deleted because:

1. It is highlighted in figure 1
2. Described in detail under the 'Methods' heading

Phase 1: Qualitative Evidence Synthesis

Comment 12: What is the research question for this systematic review?

Response: We have added a subheading '1.1 Research question and purpose' under the main heading 'Phase 1: Qualitative Evidence Synthesis'

'1.1 Research question and purpose: From the perspective of children with CP and their caregivers, what is the experience and expectation of outcomes after lower-limb surgery?' p.8

Comment 13: Study selection and criteria: the following criteria needs further clarification "(3) the study explored the experience of children and young people with CP and their family of lower limb orthopaedic surgery" it is not clear at this point what outcomes are you looking for, the experience during the surgical period?, after surgery?, the outcomes of a surgical intervention? Measured using a standardized tool/s? Themes related to goals for surgical interventions?

Response: We have adjusted the criteria to more clearly reflect the above mentioned consideration. '(3) the study explored the experience of children and young people with CP and their family following lower limb surgery and their expectations of the surgery outcomes' p.9

Comment 14: ICF linking rules – you might want to use the revised version of the linking rules published in March 2019 published in <https://www.ncbi.nlm.nih.gov/pubmed/26984720> You will need to update Table 2, there are 10 rules proposed now.

Response: Thank you for this reference. The present review was completed by January 2019, before the updated linking rules were published. However, we adjusted the interview analysis to be in line with the most recent linking rules (2019). p.10, 16 (interview)

Phase 2: Scoping Review

Comment 15: Search strategy needs revisions as key terms are missing, for example the search terms described in this section do not include the term "OUTCOME", or tools or measures, or "GAIT", you should maximize the search strategy, you might want to consultant an experienced librarian to help with this task due to the importance of this step

Response: Thank you for this point. As we updated a previous review, we maintained the original review search terms exactly the same as instructed by the Cochrane Review recommendation: 'a review update should be re-conducted using the same methods as the original review.' Therefore,

similar databases and key search terms will be replicated. Text was added under the main heading 'phase 2: Scoping Review' to clarify this point with reference to the Cochrane recommendation and the original review.

'According to the Cochrane Review recommendations, a review update should be re-conducted using the same methods as the original review,³⁵ therefore, this review was in line with the methodology of the original scoping review.⁷' p.12

Comment 16: Can you provide a rationale for conducting two systematic reviews, you could run a search for domains and outcomes including qualitative and quantitative studies.

Response: Thank you for your point. While both reviews addressed a similar overall research purpose and question with respect to their methodological differences, we felt that conducting two reviews separately is favourable for the following reasons:

1. The two reviews target two different specific perspectives which complement each other. In the Qualitative review, we aimed to look at the children and parent experiences of surgical outcomes, while the scoping review would reflect the health professionals' perspective whilst identifying outcome measures that are used.
2. The qualitative review is considered a novel part of the study, as there is no such review in the CP orthopaedic surgical literature. In this review we included all studies from database inception to date, while the scoping review is an updated version of a previous one and covers a relatively short period of time.
3. We felt that a mixed methods review would complicate the description of the methodology and findings.

Phase 3: Qualitative study

Comment 17: What is the research question that you want to answer with this study?

Response: We have added a subheading '1.1 Research question and purpose' under the main heading 'Phase 3: Qualitative study'

'3.1 Research question and purpose: The following question will guide the study: What outcome domains are considered important by ambulant children and young people with CP, their caregivers and health professionals after lower limb orthopaedic surgery?' p.14

Comment 18: Inclusion criteria: you might want to add a criteria that include "lower limb surgical intervention" as you are creating a COS for this population

Response: Table 1- stakeholder involvement criteria was updated.

'Have undergone or are being considered for lower limb orthopaedic surgical intervention' p.7

Phase 4: Delphi process

Comment 19: Participants: you are inviting children to participate in this two-stage scoring exercise, how are you planning to present the information to this group?

Response: We have adjusted the text and have provided a 2nd supplementary file including the animation, this is placed under the subheading '4.4 Delphi survey'.

'A child-friendly animation will be developed to explain the rationale of the study and promote children's understanding of the purpose of the COS (Supplementary file 2).' p.18

Comment 20: Also, the output of this project is meant to be international, are you planning to include international stakeholders? The recruitment strategy is not described for this study.

Response: We are planning to recruit an international panel. We have clarified this further in our recruitment strategies under subheading '4.3: Recruitment' and have listed relevant international stakeholders in Table 1 'Stakeholder involvement'.

'Different strategies will be followed to identify the potential panel, as shown in table 3. Clinical academics and healthcare professionals will be invited through professional societies. Specific invitations to Authors of relevant references identified through the qualitative evidence synthesis and scoping review will be targeted. Snowballing techniques will be used to ensure a representative sample of international researchers and clinicians are invited. Children and family recruitment will be initiated through the clinical care team at Nuffield Orthopaedic Centre (Oxford, UK) and will expand nationally and internationally to include patient and parent organisations and charities. p.17-18

Phase 5: Consensus meeting

Comment 21: Recruitment strategy of panel members?

And

Comment 22: International representation?

Response: We have added information about recruitment of the panel members. This is detailed under subheading '5.1 Recruitment'

'A consensus meeting will be hosted for the purpose of finalising the core outcome set. The meeting will include approximately 20 panel members. Representatives from all stakeholder groups, representing as much geographical, ethnic, demographic and cultural diversity as possible (as recommended by COMET) will be invited at this stage. Ten of those participants will be randomly selected from the Delphi survey participants and the study steering group. Face-to-face and remote access to the meeting will be available'. p.20

Comment 23: Describe the procedure for determining how outcomes will be added/combined/dropped from consideration during the consensus process

Response: We have added information about how outcomes will be finalised. This is detailed under subheading '4.4 Delphi survey'.

'At the meeting, Delphi survey results will serve as the basis for the discussion and development of the final COS to be agreed across stakeholder groups. Across all stakeholder groups, any outcome categorised as 'consensus in' will be proposed to be included in the final COS, while any outcome categorised as 'consensus out' will be excluded. The panel members will electronically vote to accept this proposal or suggest outcomes that warrant further discussion.

Outcomes that are differently categorised by different stakeholder groups and those categorised as 'no-consensus' will be discussed individually. A second round of voting will be used to agree the final COS.' p.20-21

Comment 24: Describe how missing data will be handled during the consensus process

Response: We anticipate 2 different types of missing data; first whereby a participant does not respond to the second round of the Delphi survey and second, whereby a participant does not respond to a particular item on the Delphi survey (for whatever reason). In order to minimise the risk of participants not responding to the second round of the Delphi survey we will send out regular reminder emails. In order to take into account participants who do not respond to a particular Delphi survey item we will summarise the distribution of scores and to calculate the median and interquartile range (IQR) for each Delphi survey item separately. As such the denominator for each Delphi survey item will be the number of participants completing each specific item; rather than the number of

participants completing the Delphi survey overall. We have clarified this in the text in subheading '4.4 Delphi survey' and '4.5 Analysis'.

'Each round will be open to the panel for four weeks and reminder emails will be sent at 2-week intervals in order to maximize follow up rates' p.19 (4.4. Delphi Survey)

'Descriptive statistics will be undertaken using SPSS software to summarise the distribution of scores and to calculate the median and interquartile range (IQR) for each Delphi survey item. The denominator for each Delphi survey item will be the number of participants completing that item; rather than the number of participants completing the Delphi survey overall (i.e. a participant may choose not to score a particular Delphi item for whatever reason). p.19-20 (4.5. Analysis)

Additional comments

Comment 25: Include potential limitations to the proposed studies and successful completion of proposed project

Response: A new main heading 'Discussion' was added after 'Ethics and Dissemination'.

'Strengths and limitations of this study: To our knowledge, this is the first study on the development of a core outcome set for lower limb orthopaedic surgery in ambulant children with CP. This study employed a well-established and widely used method developed by the COMET Initiative. Involving patients in COS development has become common practice to ensure the relevance of the proposed COS to all stakeholders. In this protocol, children and families will be directly engaged with the COS development through participation in the steering committee, the interviews, the Delphi process and the consensus meeting.

This study includes a comprehensive search for potentially relevant outcomes through qualitative evidence synthesis, a scoping review and interviews with stakeholders' groups. This process will be conducted by at least two independent researchers ensuring identification of all potential outcomes. This will provide a comprehensive list of all pertinent outcomes for the Delphi survey.

As the comprehensive search for outcomes focuses on the English literature and on interviews with English speaking stakeholders any outcomes available in the non-English literature may be omitted. Free-text fields will be included in the Delphi survey to allow participants to suggest any additional outcomes that they consider important. The potential imbalance between national and international participants may represent a limitation of the study. Although, the proposed COS development will aim to reach international consensus, it is possible that most participants will be recruited from the UK, which may affect the wider generalisability of the COS findings.' p.23

Comment 26: Timeline of the total project

Response: Timeline of the total project is presented in Figure 1. Separate file

Comment 27: Figure 1. Please check misspellings in the main boxes

Response: Figure 1 misspelling was corrected. Separate file

Comment 28: Sources of funding?

Response: This was revised in the manuscript as follows:

'The study is funded by the Imam Abdulrahman Bin Faisal University, Saudi Arabia as part of HA's PhD program. HD is supported by the Elizabeth Casson Trust, Health Education Thames Valley and

the Oxford Biomedical Research Centre. TT was funded by the Oxford Biomedical Research Centre.' p.24

Comment 29: ICF-CY used in first study, are you planning to use the ICF in the other studies as well?

Response: The information on ICF use in the scoping review (phase 2) and the qualitative study (phase 3) can be found in the following sections:

- Scoping review: We briefly described this in the '2.5 analysis' section- 'Content analysis will be used to identify the breadth of content of the identified outcome domains and measures, which will then be mapped to the ICF-CY domains.' p.14

- Qualitative study: We briefly described this in the '3.6 Data analysis' section- 'The list of prioritised outcomes generated by the analysis will be systematically classified using an international ICF framework linking process outlined by Cieza and colleagues.' p.16

Reviewer 2

Abstract

Comment 1: Clear presentation of the approach taken. Registration with COMET was acknowledged.

Response: Thank you for your comment.

Strengths and limitations

Comment 2: Appropriate and well-reasoned.

Response: Thank you for your comment.

Introduction

Comment 3: Population clearly defined and sufficient context provided.

Response: Thank you for your comment.

Comment 4: Acknowledgement of merits of Core Outcome Sets on research synthesis and efficiency justified.

Response: Thank you for your comment.

Comment 5: Further clarification on why this core outcome set is warranted over that cited in Page 4-5 would be beneficial. There is suggestion of a requirement for one related to 'motor function' however this message could be more clearly articulated. I am not convinced based on the current text why focusing on lower-limb orthopaedic surgery is required and the current text questions whether this is too narrow and the same COS would be relevant for pre- and post-surgery i.e. those with motor challenges. Based on the stage of this study, I would suggest that revising the text to clarify this would be beneficial. This has the potential to be really valuable and therefore please clarify/sell-this further to the reader.

Response: Thank you for this point. We have revised the text in last paragraph of the introduction.

'Lower limb orthopaedic surgery is a common intervention in the management of ambulatory children with CP. However, significant variation in the outcomes collected and reported remains a challenge. The post-surgical emotional and physical challenge that this treatment imposes on children and their families is significantly bigger than any other gait improvement intervention in this population. It is therefore of the highest importance to ensure that the surgical aims and the expected outcomes are of relevance to children and their families. Developing a COS representing all stakeholders for this specific intervention would represent the first step in this direction.' p.5

Method

Comment 6: Reporting by COS-STAP which is appreciated. Wider stakeholder group which is a huge benefit.

Response: Thank you for your comment.

Comment 7: Overall the methods are well-presented, clearly documenting and justifying the approaches taken.

Response: Thank you for your comment.

Phase 1: Qualitative Evidence Synthesis

Comment 8: I am unsure about the merit of Table 2. Could this not just be cited rather than presented in the text?

Response: We adjusted the text in the revised manuscript.

'the ICF linking rule will be used.³³ Each line of text will be coded according to its meaning and will be linked to the most precise ICF-CY code. If the content of a code not explicitly named in the ICF-CY category, the 'not defined', 'not covered', and 'health condition' category will be applied.' p.10

Phase 2: Scoping Review

Comment 9: Justification for the narrow search period (Jan 2016 to June 2019) would be helpful for the scoping review? Why has this been undertaken when important studies may been available before 2016?

Response: We have justified conducting an updating scoping review over a short period of time as follows:

'Although, previous scoping reviews for studies published between 1990 and 2015 were identified,^{7, 34} updating the review was important to ensure that recently published outcome domains and outcome measures were identified. This was important particularly because, in recent years, researchers and healthcare professionals have become more aware of patient priorities and have acknowledged the value of patient-reported outcomes.' p.12

Phase 3: Qualitative study

Comment 10: There is limited feed-forward of the work packages into the qualitative Phase 3 study. I would suggest that these findings from Phase 1 and 2 may be helpful to inform the topic guides for Phase 3. Should this be considered?

Response: We entirely agree with this point and have now addressed it under the subheading '3.5 Data collection'.

'The interview topic guide will be formed and shaped by the findings of phases 1 and 2. For example, participants will be asked to reflect on the identified outcome domains from the reviews.' p.15

Comment 11: Further detail is required in Table 1 as to who the Health Professionals are. Could the team please provide the professions they expect to approach i.e. surgeons, play therapists, physiotherapists, occupational therapists, orthotists etc etc. This would be really helpful.

Response: Further categories of health professionals were added in table 1. p.7

Phase 4: Delphi process

Comment 12: The Phase 4 Delphi is appropriately reported however please provide further information on who will be completing this, where it will be sent to, who the target recruitment sites are, what approaches are made to improve the international reach of the Delphi. This is important.

Response: Thank you for this point.

With regard to the first part of your comment, "who will be completing the Delphi survey": we have added clarification under the subheading '4.2 Selection of panel members and sample size'

'The eligibility criteria in table 1 will be used as a guide in the selection of the Delphi panel members.' p.17

With regard to the second part, "further information about recruitment": we have added a subheading '4.3 Recruitment' with the following explanation of our proposed recruitment strategies.

'Different strategies will be followed to identify the potential panel, as shown in table 3. Clinical academics and healthcare professionals will be invited through professional societies. Specific invitations to Authors of relevant references identified through the qualitative evidence synthesis and scoping review will be targeted. Snowballing techniques will be used to ensure a representative sample of international researchers and clinicians are invited. Children and family recruitment will be initiated through the clinical care team at Nuffield Orthopaedic Centre (Oxford, UK) and will expand nationally and internationally to include patient and parent organisations and charities.' p.17-18

Phase 5: Consensus meeting

Comment 13: Phase 5 needs further clarification. Phase 4 will inform what domains should be included. However the matching of domains to outcomes is a challenging process and the current text requires further detail. There is a suggestion that outcomes previously associated with the selected domains will be linked and their feasibility/psychometric properties will be evaluated against the COSMIN checklist – this process warrants its own Phase in reporting so the reader can appreciate how this is actually undertaken.

Response: Thank you for this important comment. Section 'Phase 6: Selection of outcome measures' was updated.

'Phase 6: Selection of outcome measures: After the development of a COS, it is recommended to identify a set of measurements, the "outcome measures" that would be used to evaluate the selected outcomes.⁴⁴ In order to establish a core outcome measures set, a four-step process will be followed: (1) conceptual considerations (scope); (2) identifying existing outcome measures; (3) quality assessment of the identified outcome measures and (4) generic recommendations for the selection of outcome measures for a core outcome set.

The conceptual considerations of the proposed core outcome measures set will be associated with the study scope. Accordingly, all available outcome measures used in clinical research following lower limb orthopaedic surgery in ambulant CP will be considered for the core set. An international

perspective on the subject will be captured by involving stakeholders from the study's international steering committee and consensus panel members.

Previous reviews on this field^{7, 34} alongside the scoping review (phase 2) will be used as a starting point to identify currently used outcome measures. The quality assessment of each outcome measure identified will be determined by the available systematic reviews using the Consensus-based Standards for selection of health measurement Instruments (COSMIN). This will assess the psychometric properties of outcome measures that have been used in CP clinical studies. For example, two recent systematic reviews, using a modified COSMIN method to assess gait-related outcome measures in CP will be employed to choose suitable, high-quality outcome measures.^{45, 46}

A consensus meeting with a panel of health professionals will subsequently be organised to establish appropriate outcome measures for each outcome domain identified during the COS development. Members will be asked to recommend one high-quality outcome measure per core outcome domain. If no adequate outcome measures exist for a specific core outcome, this will be acknowledged, and recommendation will be made for future development of an adequate high-quality outcome measure.' p.21-22

Comment 14: There is currently insufficient information here. Furthermore, where will the stakeholder meetings take place. Are these all in Oxford? How generalisable/transferable are the views of this group?

Response: We have added information about recruitment of the panel. This is detailed under the '5.1 Recruitment' subheading.

'A consensus meeting will be hosted for the purpose of finalising the core outcome set. The meeting will include approximately 20 panel members. Representatives from all stakeholder groups, representing as much geographical, ethnic, demographic and cultural diversity as possible (as recommended by COMET) will be invited at this stage. Ten of those participants will be randomly selected from the Delphi survey participants and the study steering group. Face-to-face and remote access to the meeting will be available'. p.20

Ethics and Dissemination

Comment 15: Appropriate

Response: Thank you for your comment.

Limitations

Comment 16: There is currently insufficient acknowledgement of the issue of generalisability/transferability of the COS internationally. The searches are based on English language publications. The consensus meetings are 'presumably' in Oxford, the Delphi reach feels UK base (but may not be). COS should be international in their importance and relevance. The current approach feels very Oxford-centric. This should be acknowledged somewhere and/or strategies highlighted to the reader as to where this has been negated.

Response:

We have clarified further in Sections 4.3 and 5.1 that the participants (both patients and health professionals) will represent international stakeholders. It is true, however, that the literature search and interviews are both in the English language only. These issues have been discussed under the 'strengths and limitations' subheading.

'Strengths and limitations: ... As the comprehensive search for outcomes focuses on the English literature and on interviews with English speaking stakeholders, any outcomes available in the non-English literature may be omitted. Free-text fields will be included in the Delphi survey to allow participants to suggest any additional outcomes that they consider important. The potential imbalance between national and international participants may represent a limitation of the study. Although, the proposed COS development will aim to reach international consensus, it is possible that most participants will be recruited from the UK, which may affect the wider generalisability of the COS findings.' p.23

VERSION 2 – REVIEW

REVIEWER	Verónica Schiariti University of Victoria, BC, Canada
REVIEW RETURNED	04-Feb-2020

GENERAL COMMENTS	Thank you for the opportunity to review the revised version of this manuscript. The authors have incorporated the recommendations raised in the first review, as such; the protocol is very informative, and the flow of the paper has significantly improved. In addition, the methodology proposed for each phase is much easier to follow. However, there are some minor comments that need to be addressed as follows:  o Figure 1. Please check misspellings in the main boxes “measurs” o References: #3 is incomplete, there is no journal o References #33 and #40 are the same reference with different numbers – please review and keep the correct citation o Please carefully review the complete list of references to meet the journal’s citation style o Table 2 - ICF linking rules – the authors incorporated the revised version of the linking rules as suggested - for the upcoming phases of the project https://www.ncbi.nlm.nih.gov/pubmed/26984720 However you need to update the content of Table 2 or change the reference #33 on this table, as I pointed out before, there are 10 rules proposed in 2019 (reference #33) but you currently show 8 rules in your table. Therefore, the paper should be considered for publication after minor revisions. Kind regards.
--

REVIEWER	Toby Smith University of East Anglia, Norwich
REVIEW RETURNED	31-Jan-2020

GENERAL COMMENTS	The authors have addressed my earlier comments. I have nothing to add.
--

VERSION 2 – AUTHOR RESPONSE

Reviewer 1

Comment 1: the protocol is very informative, and the flow of the paper has significantly improved. In addition, the methodology proposed for each phase is much easier to follow.

Response: Thank you for your comment.

Comment 2: Figure 1. Please check misspellings in the main boxes “measurs”

Response: We have corrected the typo mistake in the revised figure.

Comment 3: References: #3 is incomplete, there is no journal

Response: Thank you for pointing it out. We have corrected reference 3. p.24

Comment 4: References #33 and #40 are the same reference with different numbers – please review and keep the correct citation

Response: We have updated the reference list. p.25

Comment 5: Please carefully review the complete list of references to meet the journal's citation style

Response: We have reviewed and updated the reference list. p.24-26

Comment 6: Table 2 - ICF linking rules – the authors incorporated the revised version of the linking rules as suggested - for the upcoming phases of the project

<https://www.ncbi.nlm.nih.gov/pubmed/26984720> However you need to update the content of Table 2 or change the reference #33 on this table, as I pointed out before, there are 10 rules proposed in 2019 (reference #33) but you currently show 8 rules in your table.

Response: Thank you for your comment. We have updated Table 2 and reference belong to the analysis method of 'Phase 1: Qualitative Evidence Synthesis' . p.11

Table 2 ICF linking rules ³³

Number	Rule
1	Acquire good knowledge of the conceptual and taxonomical fundamentals of the ICF, as well as of the chapters, domains and categories of the detailed classification, including definitions before starting to link meaningful concepts to the ICF categories.
2	Identify the main concept(s) most relevant to be linked to the ICF.
3	Identify any additional concepts contained in the piece of information in addition to the main concept(s) already identified in the previous step.
4	Identify and document the perspective taken on within a certain piece of information when linking it to the ICF.
5	Identify and document the categorization of the response options.
6	Link all meaningful concepts, the most relevant and additional ones, to the most precise ICF category.
7	Use "other specified" or "unspecified" ICF categories as appropriate.
8	If the information provided by the meaningful concept is not sufficient for making a decision about the most precise ICF category, assign the concept to nd (not definable).
9	If the meaningful concept is not contained in the ICF, but is clearly a personal factor as defined in the ICF, assign the meaningful concept to pf (personal factors)
10	If the meaningful concept is not contained in the ICF, assign this meaningful concept to nc (not covered)